# Dupilumab Alters Both the Bacterial and Fungal Skin Microbiomes of Patients with Atopic Dermatitis

**DOI:** 10.3390/microorganisms12010224

**Published:** 2024-01-22

**Authors:** Naoka Umemoto, Maki Kakurai, Takanao Matsumoto, Kenta Mizuno, Otomi Cho, Takashi Sugita, Toshio Demitsu

**Affiliations:** 1Department of Dermatology, Jichi Medical University Saitama Medical Center, 1-847 Amanuma-cho, Omiya-ku, Saitama 330-8503, Japan; umenao@omiya.jichi.ac.jp (N.U.); kwackie@jichi.ac.jp (M.K.); takasaoooo@jichi.ac.jp (T.M.); k.m1.chopin@gmail.com (K.M.); demitsu@omiya.jichi.ac.jp (T.D.); 2Department of Microbiology, Meiji Pharmaceutical University, 2-522-1 Noshio, Kiyose 204-8588, Japan; chootomi@my-pharm.ac.jp

**Keywords:** atopic dermatitis, dupilumab, dysbiosis, *Malassezia*, skin microbiome, *Staphylococcus aureus*

## Abstract

The skin microbiome at lesion sites in patients with atopic dermatitis (AD) is characterized by dysbiosis. Although the administration of dupilumab, an IL-4Rα inhibitor, improves dysbiosis in the bacterial microbiome, information regarding the fungal microbiome remains limited. This study administered dupilumab to 30 patients with moderate-to-severe AD and analyzed changes in both fungal and bacterial skin microbiomes over a 12-week period. *Malassezia restricta* and *M. globosa* dominated the fungal microbiome, whereas non-*Malassezia* yeast species increased in abundance, leading to greater microbial diversity. A qPCR analysis revealed a decrease in *Malassezia* colonization following administration, with a higher reduction rate observed where the pretreatment degree of colonization was higher. A correlation was found between the group classified by the Eczema Area and Severity Index, the group categorized by the concentration of Thymus and activation-regulated chemokine, and the degree of skin colonization by *Malassezia*. Furthermore, an analysis of the bacterial microbiome also confirmed a decrease in the degree of skin colonization by the exacerbating factor *Staphylococcus aureus* and an increase in the microbial diversity of the bacterial microbiome. Our study is the first to show that dupilumab changes the community structure of the bacterial microbiome and affects the fungal microbiome in patients with AD.

## 1. Introduction

Atopic dermatitis (AD) is a chronic skin condition caused by an interplay of genetic and environmental factors, with skin barrier dysfunction being a key contributor. Filaggrin, a protein vital for keratinocyte maturation and skin barrier function, is encoded by the *FLG* gene. Patients with AD experience mutations in the *FLG* gene, leading to the inhibition of filaggrin production by the T-helper 2 cytokines interleukin (IL)-4 and IL-13 [1,2]. This leads to an increase in transepidermal water loss (TEWL) and a decrease in the water content of the stratum corneum, resulting in dry skin and a shift in the skin’s pH from slightly acidic to neutral. Thus, microbes that thrive in a neutral pH environment become prevalent on the skin of these patients. *Staphylococcus aureus*, typically absent from the skin of healthy individuals, becomes prevalent on the skin of patients with AD, leading to a reduction in the diversity of the skin bacterial microbiome [3,4,5]. The production of superantigens, including staphylococcal enterotoxins (SEA, SEB, SEC, and SED), by *S. aureus* elicits inflammatory responses that potently activate T cells in a non-specific manner [6]. Furthermore, this pathogen induces dermatitis that is Th2- and IL-17-dependent through phenol-soluble modulins (PSMs) and damages keratinocytes through alpha-toxin production. This process facilitates invasion by exogenous antigens [7]. The degree of skin colonization by *S. aureus* on the skin correlates with the severity of AD. It suggests that AD is characterized by dermatitis induced by the dysbiosis of the skin microbiome [5,8].

Topical steroids and calcineurin inhibitors are commonly used to treat AD. However, recently developed antibodies that inhibit cytokine signaling related to AD onset offer a promising alternative. The first biological medication approved for AD, dupilumab, has shown remarkable effectiveness in treating moderate-to-severe cases [9,10,11]. The medication suppresses type 2 inflammation by inhibiting the cytokine signaling of IL-4 and IL-13. Several clinical studies have shown that following the use of dupilumab, the degree of skin colonization by *S. aureus* on the skin of patients with AD decreases, and the diversity of the skin microbiome increases [12]. This indicates that the change in the skin environment—an increase in the moisture content and shift to a weak acidic pH—that accompanies the improvement in the symptoms of AD results in the correction of skin microbiome dysbiosis. In addition to dupilumab, dysbiosis can also be treated with topical steroids [13].

Besides bacteria, the skin is heavily colonized by fungi. The composition of the bacterial microbiome differs across various body sites, such as oily, moist, and dry areas. However, the fungal microbiome mostly comprises the lipophilic fungus *Malassezia*, which remains consistent across all body sites [14,15]. This major difference highlights the contrasting characteristics of the bacterial and fungal microbiomes. This microbe commonly inhabits healthy human skin, but can cause seborrheic dermatitis, or folliculitis, and aggravate AD depending on the host environment [16]. Multiple exacerbation antigens have been identified in the sera of patients with AD, and anti-Malassezia IgE antibodies have also been found [17]. Furthermore, the severity of AD is correlated with the degree of skin colonization by *Malassezia* [18,19]; the degree of *Malassezia* colonization on the skin of patients with moderate and severe cases is higher than in those with mild cases.

In recent years, the administration of dupilumab has been shown to induce dupilumab-associated head and neck dermatitis (DAHND) (also called facial and neck erythema) [20,21,22,23,24]. Although the exact cause of DAHND remains uncertain, the appearance of specific anti-*Malassezia* IgE antibodies in patients, the improvement of DAHND symptoms, and the alteration of specific antibody levels after administering the antifungal medication itraconazole indicate that *Malassezia* could play a role in the development of DAHND [25]. Because *Malassezia* could worsen AD and potentially cause DAHND, it is crucial to thoroughly investigate both the fungal and bacterial microbiomes, specifically *S. aureus*, to evaluate the correlation between the changes in the skin microbiome and the therapeutic benefits of administering dupilumab.

In this study, we examined the impact of dupilumab administration on the fungal microbiome by analyzing changes in the fungal microbiome and alterations in the degree of skin colonization by *Malassezia*, a commensal fungus. Furthermore, we investigated the correlation between the severity assessment indices of AD, such as the Eczema Area and Severity Index (EASI), Thymus and Activation-Regulated Chemokine (TARC), and the degree of skin colonization by *Malassezia*, comparing them to the bacterial microbiome. This study provides a comprehensive understanding of the skin microbiome by analyzing both bacterial and fungal microbiomes simultaneously. The findings build upon previous research on the bacterial microbiome.

## 2. Materials and Methods

### 2.1. Subjects and Sample Collection

This study enrolled participants aged 17 and above with moderate-to-severe AD who visited the Department of Dermatology at Saitama Medical Center, Jichi Medical University, during the period of November 2020 to October 2022. AD was diagnosed based on the criteria specified in the Clinical Practice Guidelines for the Management of Atopic Dermatitis 2021 [26].

Treatment was initiated with the subcutaneous introduction of 600 mg dupilumab, followed by scheduled doses of 300 mg every other week for a duration of 12 weeks. Topical steroids and anti-histamine agents were consistently used during the study. The comparative analysis included 10 healthy participants. The study protocol obtained approval from the Institutional Review Board under the approval numbers RIN S20-061 and MPU202203.

### 2.2. Scale Sample Collection and Analysis of the Skin Microbiome

Skin scales were collected from the lesion site on foreheads using OPSITE FLEXIGRID Transparent Film Dressing (6 cm × 7 cm; Smith and Nephew Medical Ltd., Hull, UK), following the method described by Sugita et al. [27]. After collection, dressings were refrigerated to extract genomic microbial DNA. Skin scales were also collected from the forehead area of healthy controls.

### 2.3. Fungal 28S rRNA Gene Sequencing and Taxonomic Assignment

The fungal microbiome was analyzed using the method described by Suzuki et al. [28]. Briefly, the D1/D2 regions of the 28S rRNA gene were amplified with the primers NL1 (5′-GCATATCAATAAGCGGAGGAAAAG-3′) and NL4 (5′-GGTCCGTGTTTCAAGACGG-3′). Pooled amplicons were prepared for sequencing using a Nextera XT DNA Library Preparation Kit (Illumina, San Diego, CA, USA) and sequenced on a MiSeq platform using MiSeq version 3 Reagent Kits (Illumina), according to the manufacturer’s instructions. As the fungal taxonomy is under revision, the latest fungal taxa were referred to from MycoBank (https://www.mycobank.org; accessed on 15 December 2023) or NCBI Taxonomy Brower (https://www.ncbi.nlm.nih.gov/taxonomy; accessed on 15 December 2023).

### 2.4. Bacterial 16S rRNA Gene Sequencing and Taxonomic Assignment

Bacterial 16S rRNA gene sequence libraries were prepared using a 16S Barcoding kit 1–24 (Oxford Nanopore Technologies, Oxford, Oxfordshire, UK; SQK-16S024) according to the manufacturer’s instructions. The full-length 16S rRNA gene was amplified using PCR with barcoded nanopore sequence primers, forward primer 27F, 5′-TTTCTGTTGGTGCTGATATTGCAGAGTTTGATCMTGGCTCAG-3′ and reverse primer, 1492R, 5-ACTTGCCTGTCGCTCTATCTTCCGGTTACCTTGTTACGACTT-3′. The amplicons were purified using Agencourt AMPure XP beads (Beckman Coulter, Brea, CA, USA) and equal amounts of amplicons per sample were pooled. The pooled samples were added to a flow cell primer kit (Oxford Nanopore Technologies, EXP-FLP002) and sequenced using a MinION Mk1C sequencer for approximately 8 h. Base calling of nanopore signals was performed using MinKNOW 22.12.5 (ONT) embedded in the Guppy version 6.4.6 pipeline (ONT). Taxonomic assignments were performed using the EPI2ME 16S workflow (ONT). The exclusion criteria for single-nanopore reads were alignment count accuracy <80%, quality score (QC) <8, and read length <1400 >1700 bp.

### 2.5. Degree of Skin Colonization by Malassezia spp. and S. aureus Determined Using qPCR

Degree of skin colonization by *Malassezia* species, *M*. *globosa*, *M*. *restricta*, and *S. aureus* were determined via qPCR using a TaqMan probe [29,30]. Amplification and detection were performed using an Applied Biosystems 7500 real-time PCR system (Applied Biosystems, Minatoku, Tokyo, Japan).

### 2.6. Statistical Analysis

Statistical analysis was performed using statistical software R (version 4.3.1, https://www.r-project.org, accessed on 1 June 2023).

## 3. Results

### 3.1. Patients

Thirty-three patients with moderate-to-severe symptoms were enrolled in this study, but three dropped out, leaving thirty patients who were observed for 12 weeks after receiving dupilumab. The analysis included their background information, such as age and sex, as well as their baseline EASI and TARC concentrations in sera at 2, 4, 8, and 12 weeks after administering dupilumab. Table 1 displays this information. The EASI reduced significantly from a mean of 27.0 ± 11.1 at baseline to 12.6 ± 8.6 after 12 weeks of administration. Furthermore, the TARC concentration in the sera notably decreased from a baseline mean of 5274 ± 5279 to 622 ± 407 pg/mL after 12 weeks of administration (Table 1).

### 3.2. Characterization of Skin Fungal Microbiome

The fungal microbiome at the lesion site in the frontal area of 30 patients was comprehensively analyzed before and at 2, 4, 8, and 12 weeks after dupilumab administration. Genera, with an average relative abundance exceeding 2%, were identified as *M. restricta*, *M. globosa*, *M. sympodialis*, *Candida parapsilosis*, *Rhodotorula mucilaginosa*, *Aspergillus*, and *Cladosporium* (Figure 1a and Appendix A, Appendix A). All nine species of *Malassezia* that have been identified to exhibit an affinity with humans (*M. restricta*, *M. globosa*, *M. sympodialis*, *M. yamatoensis*, *M. dermatis*, *M. japonica*, *M. obtusa*, *M. furfur*, and *M. slooffiae*) were found on the skin of patients. *Malassezia restricta* was the most prevalent, followed by *M. globosa*. Initially, *M. restricta* and *M. globosa* had a relative abundance of 58.1 ± 9.5% and 32.2 ± 7.8%, respectively. However, after the administration of dupilumab for 12 weeks (Figure 1a), their relative abundances decreased to 52.1 ± 6.6% and 17.9 ± 6.3%.

Fungi were classified as either *Malassezia*, non-*Malassezia* yeasts, or filamentous fungi. Figure 1b shows the relative abundances of each taxonomic category after dupilumab administration. Although the relative abundance of *Malassezia* species decreased (92.3 ± 2.5% at baseline and 8.0 ± 3.1% after 12 weeks), those of non-*Malassezia* yeasts (5.2 ± 2.3% at baseline and 16.0 ± 2.7% after 12 weeks) and filamentous fungi (2.3 ± 1.7% at baseline and 8.0 ± 3.1% after 12 weeks) increased. Among the non-*Malassezia* yeasts were *C. prapsilosis*, *Debaryomyces hansenii*, *Naganishia diffluens*, and *R. mucilaginosa*, whereas *Aureobasidium*, *Aspergillus*, *Cladosporium*, *Fusarium*, and *Penicillium* were the most prominent filamentous fungi (Appendix A and Appendix A). Although the relative abundance of the non-*Malassezia* yeasts increased with dupilumab treatment, the relative abundance of *Candida albicans* decreased with the dupilumab treatment (1.3 ± 1.2% at baseline to 0.0% ± 0.1% at 12 weeks) (Figure 1c). The relative abundance of each taxonomic category after 12 weeks of drug administration was comparable to that of the healthy individuals. The Shannon diversity index decreased over time following the administration of dupilumab (Figure 1d). Confirming the lowered abundance of *Malassezia* after dupilumab administration, the degree of skin colonization was verified using qPCR. The degree of skin colonization by *Malassezia* varies greatly among the subjects, but in all the subjects, this degree decreased after the administration of dupilumab (Figure 1e). The reduction rate was 43.1 ± 30.7%, 49.2 ± 28.0%, 47.3 ± 31.7%, and 43.9 ± 29.2% at 2, 4, 8, and 12 weeks, respectively. In addition, the degree of skin colonization by *M. restricta* and *M. globosa* also decreased (Figure 1f,g). The rate at which the *Malassezia* degree of skin colonization decreased after administering dupilumab did not exhibit a correlation with the baseline degree (*r*^2^ = 0.26, Figure 1h).

### 3.3. Characterization of the Skin Bacterial Microbiome

We also analyzed the bacterial microbiome to clarify the features of the fungal microbiome. *Staphylococcus*, *Cutibacterium*, *Corynebacterium*, *Streptococcus*, *Snodgrassella*, and *Moraxella* were the genera with an average relative abundance greater than 3%, collectively comprising approximately 80% (Figure 2a and Appendix A, Appendix A). Consistent formatting features and citation have been used. Initially, the most dominant genus was *Staphylococcus*, making up 44.3 ± 14.7%. Nevertheless, following 12 weeks of dupilumab administration, its relative abundance declined to 11.8 ± 6.1%. Subsequently, an increase from 25.4 ± 13.1% to 46.6 ± 16.6% was observed in the relative abundance of *Cutibacterium*. These two genera have an inverse correlation, as depicted in Appendix A. The relative abundance of *Corynebacterium*, *Streptococcus*, *Snodgrassella*, and *Moraxella* did not change at 12 weeks.

Figure 2b displays the Shannon microbial α-diversity. Although the baseline microbial diversification was low, it increased significantly 2 weeks after the administration of dupilumab (*p* < 0.01). The relative abundance and Shannon diversity index of the bacterial microbiome were similar between the individuals receiving dupilumab administration for 12 weeks and the healthy individuals.

The decrease in the relative abundance of the genus *Staphylococcus* following dupilumab administration allowed for the determination of the degree of skin colonization by *S. aureus* via qPCR. The baseline degree of skin colonization saw a marked reduction, exhibiting a decrease of 67.8 ± 21.6%, 86.7 ± 12.5%, 91.3 ± 11.3%, and 94.1 ± 8.4% at weeks 2, 4, 8, and 12 weeks post treatment, respectively. These results were highly significant (*p* < 0.01, Figure 2c). No *S. aureus* was detected in the control group.

### 3.4. Correlation between Degree of Skin Colonization by Malassezia and S. aureus

After the degree of skin colonization by *Malassezia* and *S. aureus* were reduced by the dupilumab treatment, we examined the correlation between the degree of the two microbes. A high correlation was indicated at baseline, and 2 and 4 weeks after administration, which were *r*^2^ = 0.77, *r*^2^ = 0.81, and *r*^2^ = 0.54, respectively. However, the correlation decreased gradually at 8 (*r*^2^ = 0.23) and 12 weeks (*r*^2^ = 0.03) after the administration, as shown in Figure 3.

### 3.5. Correlation between Clinical Score and Abundance of Skin Microbes

The baseline EASI was 27.0 ± 11.1, decreasing to 14.9 ± 9.3 after 12 weeks of dupilumab administration (Table 1). There was a weak correlation between the degree of skin colonization by *Malassezia* and EASI, with *r*^2^
*=* 0.21–0.53 (Figure 4a). Furthermore, there was a good correlation between the degree of skin colonization by *S. aureus* and EASI at all sampling times (*r*^2^ = 0.51–0.90). Both species had similar trends, with sample plots in the lower left of the graph.

It is possible to classify EASI into three severity groups. The severity of colonization was categorized as >21.1 (severe, 21.0%), 21.0–7.1 (moderate), and >7.1 (mild). The degree of *Malassezia* and *S. aureus* colonization were significantly higher in the severe group compared to the moderate and mild groups (*p* < 0.01; Figure 4b).

Out of the 30 patients in the study, 9 (30.0%) achieved EASI-75. Nonetheless, no significant distinctions were observed in the degree of skin colonization by *Malassezia* or *S. aureus* between the patients who achieved EASI-75 and those who did not (Appendix A).

### 3.6. Correlation between TARC Concentration and Abundance of Skin Microbes

The baseline concentration of TARC, a serum biomarker for AD, was 5274 ± 5279 pg/mL. After the administration of dupilumab, the concentration decreased to 622 ± 407 pg/mL after 12 weeks (Table 1). In all the figures, the sample plots were clustered on the lower left. However, no substantial correlation was observed between the degree of skin colonization by *Malassezia*, *S. aureus*, and the concentration of TARC (*r*^2^ = 0.003–0.02 for *Malassezia*, *r*^2^ = 0.003–0.05 for *S. aureus*) (Figure 5a). However, the group classified as moderate to severe, based on the TARC concentration (>700 pg/mL for moderate to severe; <700 pg/mL for mild or remission), exhibited a markedly higher degree of skin colonization compared to the *Malassezia* and *S. aureus* groups (*p* < 0.01, Figure 5b).

## 4. Discussion

Dupilumab therapy has been demonstrated to increase the diversity of the skin bacterial microbiome while decreasing *S aureus* colonization as observed in numerous clinical studies [12,31,32,33,34]. Corresponding findings were also obtained in this study. The analysis of bacterial and fungal microbiomes was a significant aspect of this study, along with the quantitative measurement of the degree of skin colonization by *S. aureus* and *Malassezia*, exacerbating factors in AD, following dupilumab administration.

Three significant findings emerged from this study. The first finding indicates that the fungal microbial composition of the skin of the patients with AD was primarily *Malassezia* at baseline, accounting for 92.3 ± 2.5% of the composition, indicating substantially low microbial diversity. However, post dupilumab administration, fungal microbiome diversity increased. At 12 weeks of dupilumab administration, the relative abundance of *Malassezia* decreased to 76.0 ± 2.9%, approaching the abundance found on healthy human skin (78.4 ± 5.2%). *Staphylococcus* and *Cutibacterium* were the predominant bacteria in the microbiome of the skin, with no exclusive colonization by any specific taxon observed. This finding emphasizes the contrast between the fungal and bacterial microbiomes. The reason behind *Malassezia’s* prevalence on the skin remains unclear. However, skin microorganisms can only colonize the skin if they can produce secretory lipase, which can hydrolyze sebum, their primary source of carbon and nitrogen [16]. *Malassezia* has a greater number of lipase genes in its genome than other fungi [35]. *Malassezia restricta* and *M. globosa* are the most prevalent in cases of *Malassezia*-related skin conditions, including AD, and seborrheic dermatitis [27,36]. Following administration, the relative abundances of *M. restricta* and *M. globosa* decreased gradually (Appendix A), whereas other species of *Malassezia* became more abundant. This study found an increase in *Malassezia* diversity during the treatment, with a corresponding improvement in the diversity of the fungal microbiome, which was attributed to non-*Malassezia* yeast species. Despite their low abundance, these non-*Malassezia* yeast species showed the greatest increase after the dupilumab administration (Appendix A). In dermatitis, where resident skin microbes are causative or exacerbating factors, the microbial diversity decreases and then increases as symptoms improve. The mechanism for the decrease in skin colonization by *Malassezia* and the increase in the diversity of the fungal microbiome following the administration of dupilumab is likely similar to that of the bacterial microbiome. As symptoms improve, the skin’s pH shifts from neutral to weakly acidic. While *S. aureus* prefers a neutral environment, coagulase-negative Streptococci (CNS) have reduced growth in this environment, leading to dysbiosis [3]. In a weakly acidic environment, *S. aureus* cannot grow, whereas CNS prefer this environment, resulting in an increase in the diversity of the bacterial microbiome. Although *Malassezia* can grow in both neutral and weak acidic environments, it prefers a neutral environment. The fact that the fungal microbiome of the AD patients becomes closer to that of the healthy individuals following the administration of dupilumab suggests that fungi other than *Malassezia* also prefer a weakly acidic environment.

The abundance of C. albicans gradually decreased post administration. Although a common resident microbe on healthy human skin, it is infrequently isolated from it. Despite this, *C. albicans* can cause opportunistic infections in the organs and mucous membranes of immunocompromised patients. Reduced defensin expression on the skin of patients with AD could trigger the growth of *C. albicans*. Specific IgE antibodies against this organism are produced in the patient’s serum, which indicates it as an aggravating factor of AD [37,38]. Filamentous fungi such as *Aspergillus* and *Cladosporium* were also found but considered of environmental origin. The degree of skin colonization by *Malassezia* remained mostly unchanged after the fourth week of dupilumab treatment. The degree of skin colonization was reduced to 49.2% ± 28.0% at 4 weeks and 43.9 ± 29.2% at 12 weeks, compared to the baseline. Following the administration, the degree of skin colonization *by Malassezia* was almost identical to that of healthy individuals. This contrasts with *S. aureus*, which vanished almost completely by week 12. This difference may be because *S. aureus* is rarely found in healthy individuals, whereas *Malassezia* is a commensal microorganism in humans. Therefore, the degree of skin colonization by *Malassezia* and *S. aureus* was highly correlated up to two weeks after administration, but the correlation decreased thereafter. The severity of AD is correlated with the degree of skin colonization by *Malassezia* [18,39]. In this study, the degree of skin colonization by *Malassezia* in patients with moderate-to-severe AD approximated the microbiome state of healthy individuals after dupilumab therapy. The third finding demonstrated a significant correlation between the degree of skin colonization by *Malassezia* and the EASI. *Malassezia* serves as an exacerbating factor in AD, and the level of anti-*Malassezia* specific antibodies found in the patient’s serum correlates with the severity of the disease—whether the patient has a severe, moderate, or mild condition [18]. The correlation between the degree of skin colonization by *S. aureus* and EASI is well established [12]. This study further corroborates the notion that the degree of skin colonization by these two microorganisms indicates the clinical symptoms of AD.

The superantigen of *S. aureus* increases the expression [40]. Therefore, there should be a correlation between TARC concentration and AD severity. However, no strong correlation was observed between the degree of skin colonization by *S. aureus*, *Malassezia*, and TARC concentrations in our study. Callewaert et al. [12] demonstrated a significant correlation between baseline *S. aureus* colonization and TARC concentration. This correlation might be because of individual patient backgrounds, although detailed speculation was impossible. However, a difference in *S. aureus* colonization was observed between the groups with moderate to severe (>700 pg/mL) and mild (<700 pg/mL) TARC concentrations, suggesting that TARC may not reflect the microbiome composition as sensitively as EASI. Further research must clarify the relationship between TARC and the microbiome.

Dupilumab demonstrates remarkable therapeutic effects in moderate-to-severe AD. However, in 10% of the patients who receive this medication, DAHND, commonly known as facial and neck erythema, has emerged as a clinical issue [20]. The presence of anti-*Malassezia*-specified antibodies in the serum of the affected patients, coupled with the disappearance of DAHND symptoms following antifungal agent administration, suggests the involvement of *Malassezia* in the development of this condition [19]. *Malassezia* is found everywhere in the body except for the soles, but is most concentrated in the head and neck. Patients with head and neck type (HNAD) exhibit greater levels of anti-*Malassezia*-specific antibodies in their serum compared to those without HNAD, resulting in the prior use of antifungal treatments for HNAD [19]. Although DAHND is a drug-induced dermatitis and is not equivalent to HNAD, the efficacy of antifungal treatments for both conditions is intriguing. In our trial of 30 patients with AD, no cases of DAHND were found. Although *Malassezia’s* role in the onset of DAHND has been suggested, the exact mechanism remains uncertain. Measurement of both anti-*Malassezia* antibodies and the degree of skin colonization can establish *Malassezia* as a biomarker for DAHND onset.

This study’s strength lies in its novel simultaneous analysis of fungal and bacterial microbiomes post administration of dupilumab. Prior research on the relationship between AD and skin microbiota has predominantly centered on *S. aureus*, but this investigation exposed alterations in the fungal microbiome following AD symptom improvement, highlighting bacteria and fungi interactions on the skin [15]. The correlation between clinical scores and colonization by *S. aureus* and *Malassezia* in patients with AD showed that *S. aureus* had a more significant impact on the disease severity than *Malassezia*. Recent research has indicated that *Malassezia* can play a substantial role in controlling *S. aureus*. On the skin, microorganisms create biofilms and depend on protein A (SpA) for their formation. However, secreted aspartyl proteinases from *Malassezia* hydrolyze SpA, inhibiting biofilm formation. This suggests that *Malassezia* interacts with *S. aureus* to suppress inflammatory responses [41,42].

In environments with high levels of *S. aureus*, it is probable that the insufficient production of Malassezia SAP occurs to impede biofilm formation. Whereas *Malassezia* exacerbates AD in patients, it also aids in the eradication of *S. aureus*. This investigation offers valuable insights into the interplay of these microorganisms.

A limitation of the study is that the analyses were performed only on the forehead, which was chosen for efficient detection due to the highest microbial colonization occurring on the head. The analysis of other areas such as the abdomen, back, and limbs could enable a comparison of the degree of skin colonization, particularly in DAHND-affected regions. Measuring the levels of *Malassezia* colonization and serum anti-*Malassezia* specific antibodies after dupilumab administration could more effectively demonstrate its microbiological therapeutic effects. Furthermore, the EASI-75 achievement rate was 30%, which is lower than the rates reported in other studies. Conducting future trials with larger sample sizes would improve data robustness.

## 5. Conclusions

This study shows that the administration of dupilumab significantly modifies the fungal and bacterial microbiomes in patients with AD. This reduces the colonization of *Malassezia*, specifically *M. restricta* and *M. globosa*, and decreases the presence of *S. aureus*. Thus, this produces an increased microbial diversity closer to healthy individuals’ microbiomes. During treatment, the initial correlation between *Malassezia* and colonization with *S. aureus* weakens. A noteworthy correlation was found between the microbial changes and the clinical severity of AD, as measured by the EASI. Our findings present evidence that dupilumab therapy not only ameliorates clinical symptoms in AD but also fosters a beneficial impact on the skin microbiome. This highlights the intricate interplay between treatment, microbial colonization, and disease severity in AD.

## Figures and Tables

**Figure 1 microorganisms-12-00224-f001:**
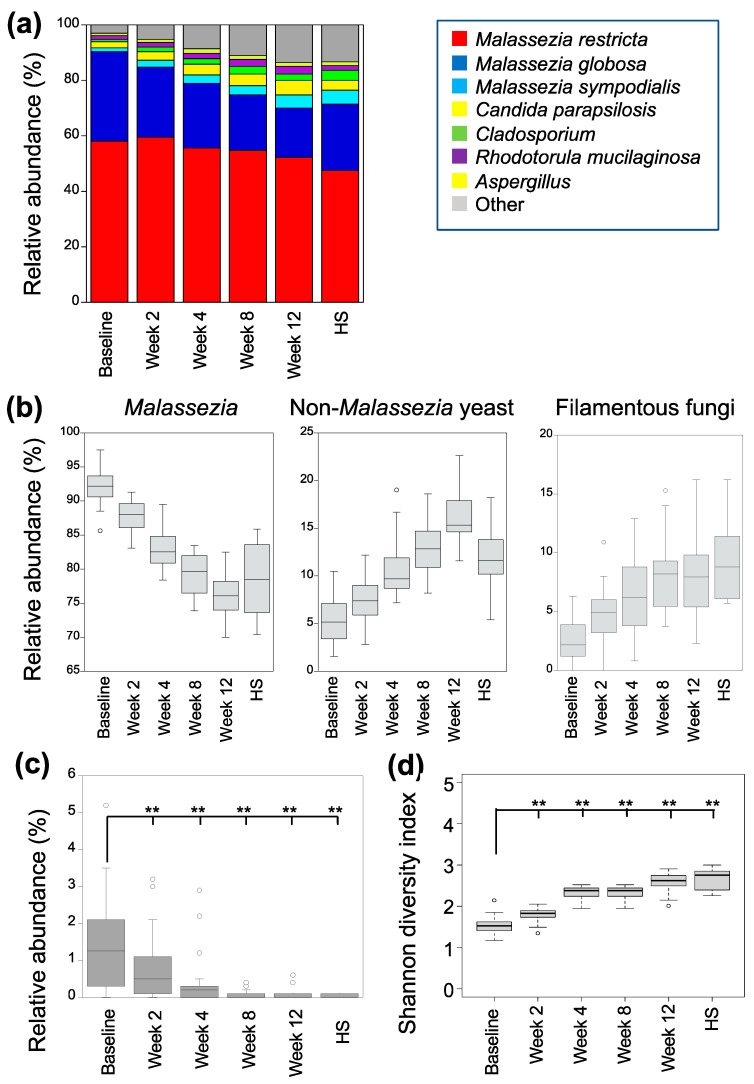
Changes in the fungal microbiome after dupilumab administration. Thirty lesion sites on the foreheads of patients were analyzed. (**a**) Abundant fungal genera. Genera with a relative abundance of >2% are shown. The relative abundance of *Malassezia restricta* decreased after dupilumab administration, whereas that of *M. globosa* increased. (**b**) Relative abundance of three taxonomic categories. The relative abundances of the *Malassezia* species were lower than those of the non-*Malassezia* yeasts and filamentous fungi. (**c**) Relative abundance of *Candida albicans*, ** *p* < 0.01. (**d**) Shannon diversity of the bacterial microbiome. Diversity increased 2 weeks after dupilumab administration. ** *p* < 0.01. (**e**–**g**) Changes in degree of skin colonization by *Malassezia* revealed using qPCR. (**e**) Overall *Malassezia* species. (**f**) *M. restricta.* (**g**) *M. globosa*. The degree of skin colonization by overall *Malassezia*, *M. restricta*, *and M. globosa* decreased compared to baseline. Experiments were performed in triplicate for each sample, ** *p* < 0.01. (**h**) Correlation between reduction rate related to baseline and degree of skin colonization by *Malassezia* species (*r*^2^ = 0.26). HS, healthy subjects.

**Figure 2 microorganisms-12-00224-f002:**
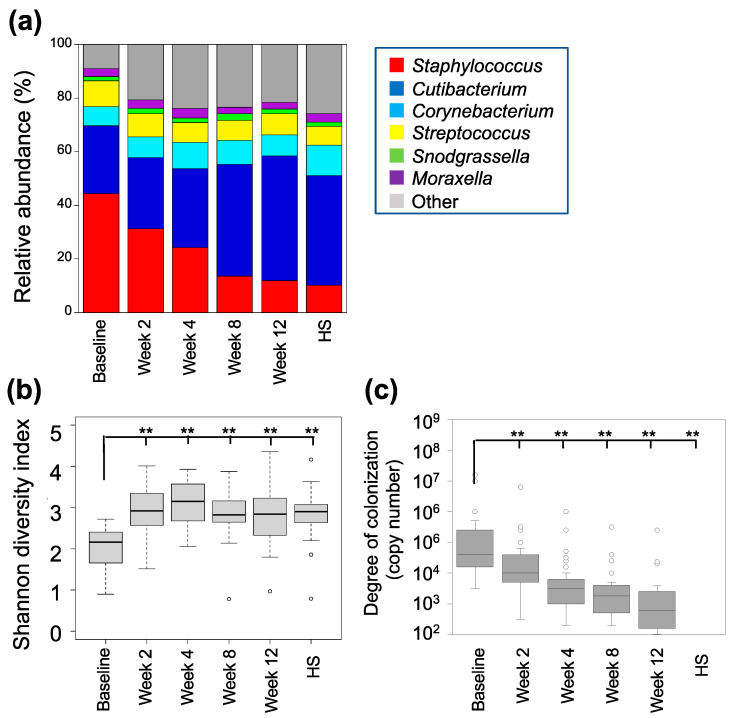
Changes in the bacterial microbiome after dupilumab administration. Thirty lesion sites on the foreheads of patients were analyzed. (**a**) Abundant bacterial genera. Genera with a relative abundance of >2% are shown. The relative abundance of *Staphylococcus* decreased after dupilumab administration, whereas that of *Cutibacterium* increased. (**b**) Shannon diversity of the bacterial microbiome. Diversity increased 2 weeks after dupilumab administration. ** *p* < 0.01. (**c**) Changes in degree of skin colonization by *Staphylococcus aureus* revealed using qPCR. Experiments were performed in triplicate for each sample. The degree of skin colonization decreased compared to that at baseline. ** *p* < 0.01. HS, healthy subjects.

**Figure 3 microorganisms-12-00224-f003:**
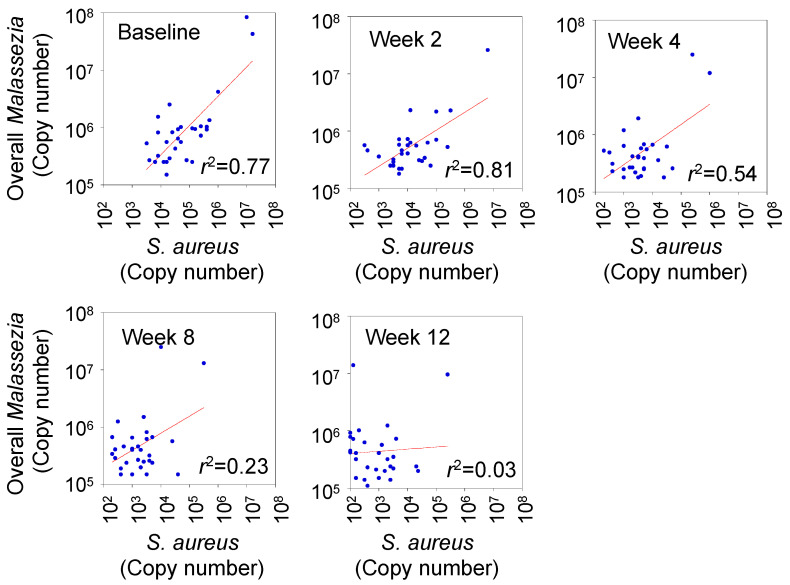
Relationship between degree of skin colonization by *Staphylococcus aureus* and *Malassezia* after dupilumab therapy. Degree of skin colonization at baseline and at 2, 4, 8, and 12 weeks after administration were analyzed.

**Figure 4 microorganisms-12-00224-f004:**
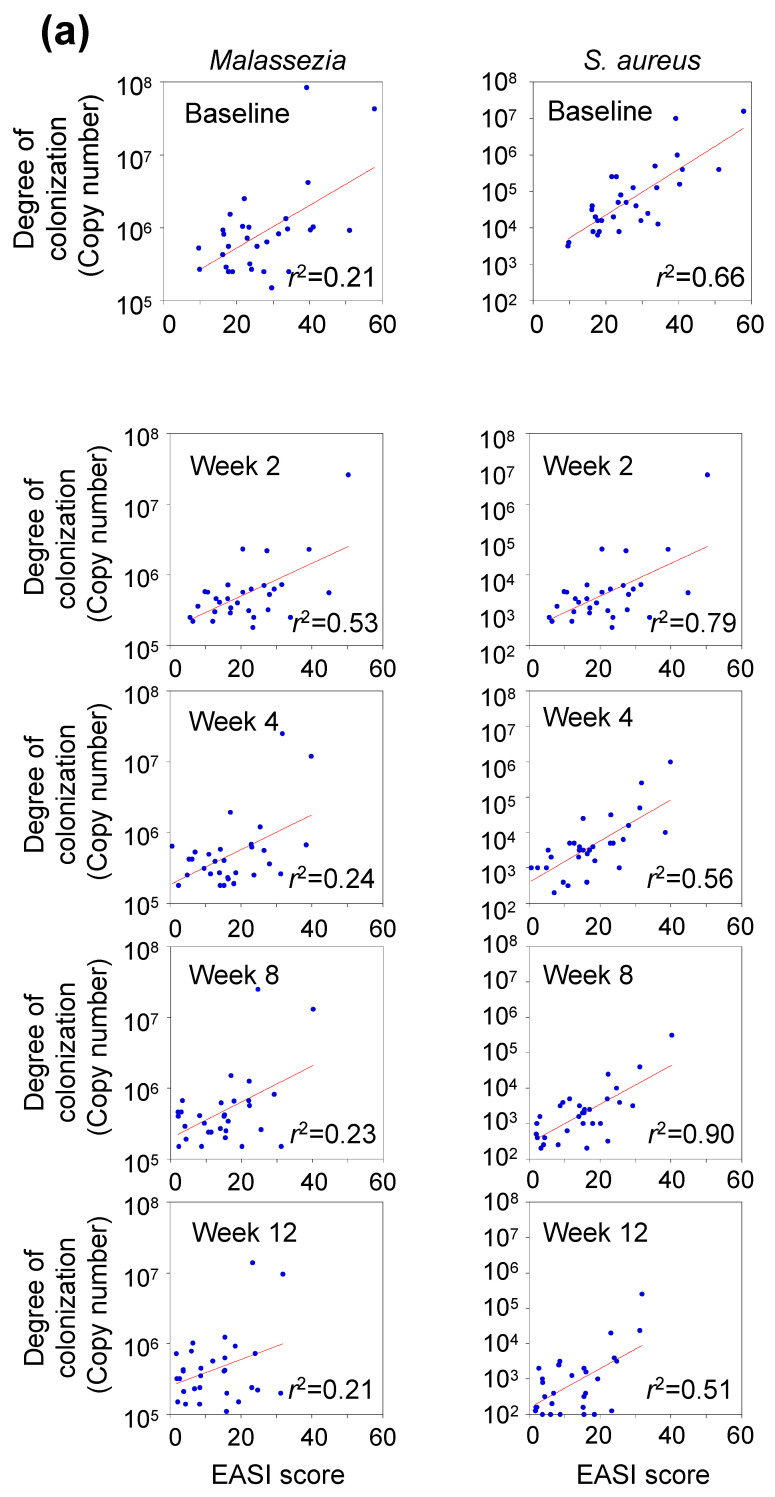
Relationship between EASI and degree of skin colonization by *Malassezia* or *Staphylococcus aureus.* (**a**) Temporal changes in the correlation between EASI scores and degree of skin colonization by *Malassezia* and *S. aureus* after dupilumab administration. Data were calculated from all sampling points (n = 150). (**b**) Degree of skin colonization by *S. aureus* and *Malassezia* based on EASI severity groups. Data were calculated from all sampling points (n = 150). ** *p* < 0.01; NS, not significant; Mann–Whitney U test. EASI, Eczema Area and Severity Index.

**Figure 5 microorganisms-12-00224-f005:**
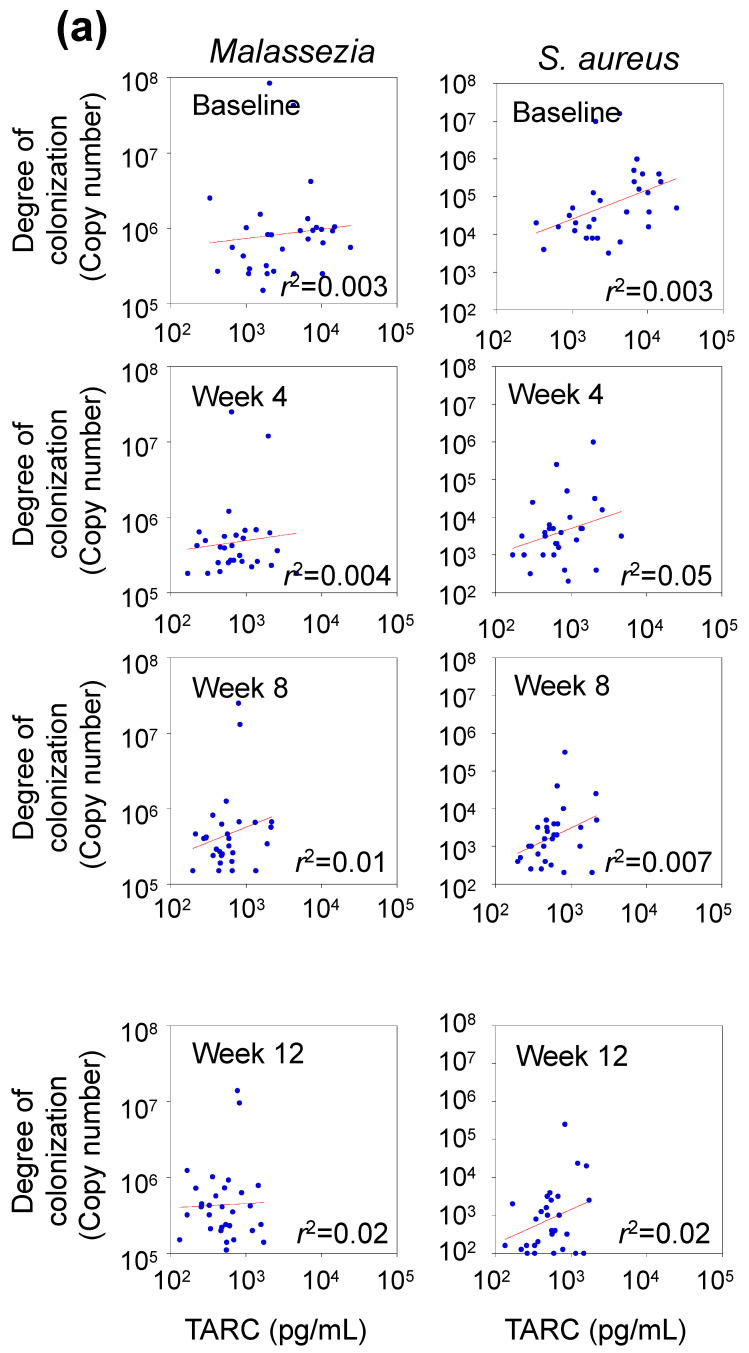
Relationship between serum TARC concentration and degree of skin colonization by *Malassezia* or *Staphylococcus aureus.* Measurements were not conducted 2 weeks after dupilumab administration for insurance reasons. TARC and thymus and activation-regulated chemokine levels. (**a**) Temporal changes in the correlation between serum TARC concentrations and degree of skin colonization by *Malassezia* and *S. aureus* after dupilumab administration. Data were calculated for all sampling points (n = 119). (**b**) Degree of skin colonization by *Malassezia* and *S. aureus* based on serum TARC concentration severity groups. Data were calculated for all sampling points (n = 119). ** *p* < 0.01; Mann–Whitney U test.

**Table 1 microorganisms-12-00224-t001:** Characteristics of the patients involved in this study.

Characteristics	Patients with Atopic Dermatitis					Healthy Individuals
	Baseline	Follow-Up				
		Week 2	Week 4	Week 8	Week 12	
Age,	42.1 ± 13.2					40.9 ± 12.0
mean + SD (range)	(17–64)					(21–60)
Female sex no. (%)	10 (33%)					2 (20%)
EASI score,	27.0 ± 11.1	21.6 ± 10.6	17.3 ± 9.8	14.9 ± 9.3	12.6 ± 8.6	
mean + SD (range)	(9.6–57.8)	(5.6–50.3)	(0.5–39.8)	(2.0–40.2)	(1.7–31.8)	
Chamge in EASI score from baseline,%		21.2	36.4	47	53.3	
EASI-75, no. (%)		0 (0%)	3 (10%)	5 (16.7%)	9 (30%)	
Serum total IgE,	13,875 ± 8920				7883 ± 4042	
mean + SD (range)	(610–25,000)				(210–19,000)	
TARC (pg/mL),	5274 ± 5279		1002 ± 910	715 ± 526	622 ± 407	
mean + SD (range)	(331–24,200)		(168–4656)	(196–2176)	(132–1722)	

SD, standard deviation.

## Data Availability

Data are contained within the article and Appendix A.

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
