# Peer review of "Dupilumab Alters Both the Bacterial and Fungal Skin Microbiomes of Patients with Atopic Dermatitis"

_microorganisms, 2024, doi:10.3390/microorganisms12010224_

Round 1

Reviewer 1 Report

Comments and Suggestions for Authors

Dear Authors,

The manuscript contains many passages in which some words used are not part of the English or American linguistic dictionary. Once again I found our manuscript in Microorganisms 2021, 9, 2132. https://doi.org/10.3390/ microorganisms9102132 that appears very similar to this.

Comments on the Quality of English Language

The manuscript shows many passages in which some words used are not part of the English linguistic dictionary. The author could attempt to modify the text copied from other authors to prevent the manuscript from being discovered.

Author Response

Response to Reviewer’s comments:

Reviewer 1

Question 1: The manuscript contains many passages in which some words used are not part of the English or American linguistic dictionary. Once again I found our manuscript in Microorganisms 2021, 9, 2132. https://doi.org/10.3390/ microorganisms9102132 that appears very similar to this.

The manuscript shows many passages in which some words used are not part of the English linguistic dictionary. The author could attempt to modify the text copied from other authors to prevent the manuscript from being discovered.

Response:

Thank you for your valuable and considerate comments on our submission. To address your concerns on language, we would like to highlight that this manuscript has undergone English language editing by a native expert (Editage, https://www.editage.jp). Additionally, the overall similarity score checked using iThenticate was 3%, leading us to believe that there has been no direct copying from other papers.

Reviewer 2 Report

Comments and Suggestions for Authors

Review for manuscript assigned as: microorganisms-2765285 (Manuscript ID)

Dear Authors,

Your findings present evidence that dupilumab demonstrates remarkable therapeutic effects and therapy based on this drug not only significantly reduces clinical symptoms in patients with AD but also has a beneficial impact on the skin microbiome. In my opinion your work is very interesting and has not only cognitive, but also an exceptionally applied character. This paper contributes a lot to dermatology, medical mycology, therapy of AD, pharmacology and finally for better understanding of the microbiome. Study performed by Authors allow to show new possibilities in the treatment of atopic dermatitis and the benefits of application of dupilumab. The Authors put a lot of work into obtaining valuable results, preparing them meticulously and presenting them in a very accessible way. Nevertheless, some minor shortcomings could not be avoided, which is completely understandable.

All the figures and tables are appropriate for this type of article. In general, the paper has a logical flow. The abstract well correspond with the main aspects of the work and the literature is well selected for well prepared "Introduction" and "Discussion" however, it could be improved to some extent what I described below.

As a reviewer I am obligated to pay attention even to less important weak points of this work and all mentioned below comments should be carefully considered. All the comments below do not diminish my high assessment of this work.

Title

In my humble opinion ,,Dupilumab Alters both the Bacterial and Fungal Skin Microbiomes of Patients with Atopic Dermatitis” sounds better.

Abstract

Line 16

"...both fungal and bacterial skin microbiomes..." sounds better and is more correct

Introduction

Lines 64-73

To the best of my knowledge, there are other publications dealing with the issues described here that have been omitted and may be an important source of information for potential readers, giving a comprehensive look at the problems described (for example: doi: 10.1111/bjd.21019; doi:10.1007/s40257-021-00646-z; doi: 10.1111/dth.15140; doi: 10.3390/life12020299). The Authors did not refer to these works in the ,,Discussion”.

Line 79

Malassezia spp. are not only ,,commensal fungi” but also opportunistic pathogens

Materials and Methods

Line 102

,,...to extract genomic microbial DNA...” sounds better

Line 111

Current names of the fungal taxa you can also check on Taxonomy Browser (NCBI) link: Taxonomy browser (Encephalitozoon cuniculi) (nih.gov)

Line 118

There shouldn't be any space in sequence of forward primer 27F

Lines 129 and 130

,,Degree of skin colonization by Malassezia spp. and S. aureus determined using qPCR” sounds better and more appropriate than ,,Colonization level by Malassezia and S. aureus using qPCR”

Similarly, ,,Degree of skin colonization by Malassezia species...” is more correct than ,,Colonization levels of Malassezia species...”

Results

Lines 159-161

Authors wrote (quote) ,,...after the administration of dupilumab for 12 weeks (Figure 1A), their (Malassezia) relative abundances decreased...”.

The interesting result obtained could have a better impact for the audience if in the "Introduction" the Authors referred to the antifungal or biological activity of Dupilumab. Has anyone tested the antifungal activity of this compound so far? Is there any information about it in the literature? Is there any information in the literature about the inhibitory activity of this compound (used solely or in combination) toward Malassezia? In my opinion, this would be valuable information that would show in a better light the essence of the interesting results obtained by Authors.

Line 168

To the best of my knowledge should be ,,Debaryomyces hansenii” (correct name)

Lines 190-191 , Figure 1

In the legend (description) to Figure 1, the Authors wrote (I quote): "The relative abundance of Malassezia restricta decreased after dupilumab administration, whereas that of M. globosa increased". This is not true in relation to what is shown in the attached Figure 1. Analysis of the relative abundance allows us to conclude that it is exactly the opposite, i.e. the relative abundance of Malassezia restricta increased, whereas that of M. globosa decreased. In my opinion, it is exactly as described by the Authors earlier in the text (lines 179-182).

Figure 1

The letter "L" is written everywhere using bold - correction needed.

Line 193, Figure 1

Should be ,,Candida albicans

Lines 193-194, Figure 1

According to what is shown on Figure 1 "Relative abundance of Candia albicans" should be assigned as "d" and "Shannon diversity of the bacterial microbiome" should be assigned as "c". There is still a question to be clarified, namely whether it should be "Shannon diversity of the bacterial microbiome" or "Shannon diversity of the fungal microbiome"

Line 204

Why did for fungi Authors use the criterion "relative abundance greater than 2%" whereas for bacteria "relative abundance greater than 3%" ? I think it should be explained in the text (maybe within Materials and methods)

Line 206-208

It should be better in the substantive context if the Authors determined which specific species of the Staphylococcus genus were dominant and the decrease of which was observed after 12 weeks. There is a significant difference between the presence of S. aureus and S. epidermidis on the skin. The latter is more typical colonizer and component of the human skin microbiome. Do Authors have such data to refer to them more precisely here. This would be very valuable information.

Lines 218-222

Authors wrote (quote) ,,The decrease in the relative abundance of the genus Staphylococcus following dupilumab administration....”

The interesting results obtained could have a better impact for the audience if in the "Introduction" or ,,Discussion” the Authors referred to the antibacterial activity of Dupilumab. Has anyone tested the antibacterial activity of this compound so far? Is there any information about it in the literature? Is there any information in the literature about the inhibitory activity of this compound (used solely or in combination) toward Staphylococcus aureus? In my opinion, this could be valuable information that would show in a better light the essence of the interesting results obtained by Authors. In my opinion, it would be good if the Authors referred in the "Introduction" or in the "Discussion" to the following articles (doi:10.1016/j.jid.2019.05.024; doi: 10.1111/j.1365-2133.2006.07410.x; doi: 10.1016/j.jaci.2018.08.022). All these papers deal with the effect of Dupilumab on Staphylococcus.

Figure 2

The letter "L" is written everywhere using bold - correction needed.

Line 247

Instead of ,,Malassezia restricata” should be ,,Malassezia restricta

Discussion

Lines 299-301

On the discussion pages, the Authors draw the following conclusion (I quote) ,,This indicates that improving the diversity of the skin bacterial microbiome and eliminating S. aureus could serve as microbiological markers indicating successful AD treatment”. Isn't this the Authors' conclusion too far-reaching and subjective? What are currently officially recognized criteria by dermatologists or organizations specializing in atopic dermatitis that could be recognized as microbiological markers indicating successful AD treatment. Do we currently have sufficient knowledge on this subject?

Line 316

,,...their primary source of carbon and nitrogen” sounds better and more completely

Lines 317-319

I can agree that M. restricta and M. globosa are the most prevalent as the factors indirectly related with AD and seborrheic dermatitis, but I can't agree with the statement that these are the most prevalent etiological factors of pityriasis versicolor.

Lines 319-320

On the discussion pages Authors wrote (I quote) ,,Following administration, the relative abundances of M. restricta and M. globosa decreased gradually”. How does this relate to the previous statement (I quote) and the results presented in Figure 1? ,,Furthermore, whereas the colonization levels of M. restricta increased by approximately 10% from 62.0 ± 11.0% at baseline to 72.0 ± 11.0%, the colonization levels of M. globosa decreased from 34.6 ± 8.4% 181 to 24.6 ± 8.4% (Figures 1F and 1G) – lines: 179-182. As I see there is discrepancy.

Line 331

To the best of my knowledge Candida albicans can cause superficial infections of the skin, nails and mucous membranes not only in case of immunocompromised patients.

Author Response

Response to Reviewer’s comments:

Reviewer 2

Dear Authors, 

Your findings present evidence that dupilumab demonstrates remarkable therapeutic effects and therapy based on this drug not only significantly reduces clinical symptoms in patients with AD but also has a beneficial impact on the skin microbiome. In my opinion your work is very interesting and has not only cognitive, but also an exceptionally applied character. This paper contributes a lot to dermatology, medical mycology, therapy of AD, pharmacology and finally for better understanding of the microbiome. Study performed by Authors allow to show new possibilities in the treatment of atopic dermatitis and the benefits of application of dupilumab. The Authors put a lot of work into obtaining valuable results, preparing them meticulously and presenting them in a very accessible way. Nevertheless, some minor shortcomings could not be avoided, which is completely understandable. 

All the figures and tables are appropriate for this type of article. In general, the paper has a logical flow. The abstract well correspond with the main aspects of the work and the literature is well selected for well prepared "Introduction" and "Discussion" however, it could be improved to some extent what I described below.

As a reviewer I am obligated to pay attention even to less important weak points of this work and all mentioned below comments should be carefully considered. All the comments below do not diminish my high assessment of this work.

Response:

We greatly appreciate the time and effort you have put into carefully reviewing our manuscript, and are thankful for your valuable suggestions and comments. We are delighted to learn that the significance and potential contributions of our work have been acknowledged. We understand that the paper can be improved on several aspects and are grateful for your kind assessment and guidance. Accordingly, we have addressed each of your comments below and modified the manuscript as per your suggestions.

Question 1: Title

In my humble opinion, Dupilumab Alters both the Bacterial and Fungal Skin Microbiomes of Patients with Atopic Dermatitis” sounds better.

Response:

Thank you for this suggestion. We have modified the title of the manuscript accordingly. “Dupilumab Alters both the Bacterial and Fungal Skin Microbiomes of Patients with Atopic Dermatitis”

Question 2: Abstract

Line 16

"...both fungal and bacterial skin microbiomes..." sounds better and is more correct

Response:

We have modified this sentence accordingly in the revised manuscript (line 16):

“….both fungal and bacterial skin microbiomes….”

Question 3: Introduction

Lines 64-73

To the best of my knowledge, there are other publications dealing with the issues described here that have been omitted and may be an important source of information for potential readers, giving a comprehensive look at the problems described (for example: doi: 10.1111/bjd.21019; doi:10.1007/s40257-021-00646-z; doi: 10.1111/dth.15140; doi: 10.3390/life12020299). The Authors did not refer to these works in the “Discussion”.

Response:

We thank the reviewer for this suggestion. After administering dupilumab, the appearance of dupilumab-associated head and neck dermatitis (DAHND) in the head and neck regions of patients was originally cited, so we believed these citations to be sufficient. However, we agree that adding your suggested references could further improve reader comprehension. Therefore, we have included the following three papers as references 22–24 in the revised manuscript. Please also note that doi: 10.3390/life12020299 has already been cited as reference 12.

  1. Kozera, E.; Stewart, T.; Gill, K.; De La Vega, M.A.; Frew, J.W. Dupilumab-associated head and neck dermatitis is associated with elevated pretreatment serum Malassezia-specific IgE: a multicentre, prospective cohort study. Br. J. Dermatol. 2022, 186, 1050-1052.DOI: 10.1111/bjd.21019
  2. Muzumdar, S.; Skudalski, L.; Sharp, K.; Waldman, R.A. Dupilumab facial redness/dupilumab facial dermatitis: a guide for clinicians. Am. J. Clin. Dermatol. 2022, 23, 61-67.DOI: 10.1007/s40257-021-00646-z
  3. Ordóñez-Rubiano, M.F.; Casas, M.; Balaguera-Orjuela, V. Dupilumab facial redness: clinical characteristics and proposed treatment in a cohort. Dermatol. Ther. 2021, 34, e15140. DOI:10.1111/dth.15140

Question 4: Line 79

Malassezia spp. are not only “commensal fungi” but also opportunistic pathogens

Response:

As you rightly pointed out, Malassezia can cause infections in immunocompromised hosts and act as causative agents or exacerbating factors in seborrheic and atopic dermatitis. However, as Malassezia are also commensal fungi, we wish to retain our current description in the manuscript. We appreciate your understanding.

Question 5: Materials and Methods

Line 102

“...to extract genomic microbial DNA...” sounds better

Response:

We have modified this sentence according to your suggestion, as follows (line 109): “...to extract genomic microbial DNA...”

Question 6: Line 111

Current names of the fungal taxa you can also check on Taxonomy Browser (NCBI) link: Taxonomy browser (Encephalitozoon cuniculi) (nih.gov)

Response:

As per your suggestion, fungal taxa have been referred from the NCBI Taxonomy Browser (https://www.ncbi.nlm.nih.gov/taxonomy), in addition to Mycobank (lines 120-121).

Question 7: Line 118

There shouldn't be any space in sequence of forward primer 27F

Response:

Thank you for bringing this to our attention. The space in Primer 27 sequence has now been removed in accordance with your suggestion (line 127).

Question 8: Lines 129 and 130

“Degree of skin colonization by Malassezia spp. and S. aureus determined using qPCR” sounds better and more appropriate than “Colonization level by Malassezia and S. aureus using qPCR”

Similarly, “Degree of skin colonization by Malassezia species…” is more correct than “Colonization levels of Malassezia species...”

Response:

In accordance with your suggestion, we have modified “colonization level of Malassezia” to “degree of skin colonization by Malassezia” in the revised manuscript.

Question 9: Results

Lines 159-161

Authors wrote (quote) “...after the administration of dupilumab for 12 weeks (Figure 1A), their (Malassezia) relative abundances decreased...”. 

The interesting result obtained could have a better impact for the audience if in the "Introduction" the Authors referred to the antifungal or biological activity of Dupilumab. Has anyone tested the antifungal activity of this compound so far? Is there any information about it in the literature? Is there any information in the literature about the inhibitory activity of this compound (used solely or in combination) toward Malassezia? In my opinion, this would be valuable information that would show in a better light the essence of the interesting results obtained by Authors.

Response:

We thank the reviewer for this detailed suggestion. To clarify, we have not investigated the direct antifungal effect of dupilumab against Malassezia, nor has it been reported in previous literature. Dupilumab is a recombinant human IgG4 monoclonal antibody against the α subunit of the human interleukin-4 receptor, so it is reasonable to assume that it does not exhibit antifungal activity. Administration of dupilumab improves the skin environment—it increases moisture content and shifts skin pH from neutral to weak acidic—which is the natural environment of healthy skin. The fact that the degree of Malassezia colonization on the skin of patients with atopic dermatitis approaches that of healthy individuals, and the diversity improves following administration of dupilumab, should be considered a result of the improvement of dysbiosis and not attributed to the antifungal action of dupilumab. Also, please refer to our response to Question 17 below.

Question 10

Line 168

To the best of my knowledge should be “Debaryomyces hansenii” (correct name)

Response:

We have corrected the typographical error from “Debaryomyces hanseni” to “Debaryomyces hansenii”, accroding to your indication (line 177).

Question 11

Lines 190-191, Figure 1

In the legend (description) to Figure 1, the Authors wrote (I quote): "The relative abundance of Malassezia restricta decreased after dupilumab administration, whereas that of M. globosa increased". This is not true in relation to what is shown in the attached Figure 1. Analysis of the relative abundance allows us to conclude that it is exactly the opposite, i.e. the relative abundance of Malassezia restricta increased, whereas that of M. globosa decreased. In my opinion, it is exactly as described by the Authors earlier in the text (lines 179-182).

Response:

We apologize for this inconvenience and acknowledge that our previous statement was incorrect. We have made the following corrections in the revised manuscript (line @@): 

“The baseline degree of skin colonization by Malassezia was observed to decrease over time after administration of dupilumab, with reduction rates of 43.1 ± 30.7%, 49.2 ± 28.0%, 47.3 ± 31.7%, and 43.9 ± 29.2% at 2, 4, 8, and 12 weeks, respectively (Figure 1E). In addition, the degrees of skin colonization by M. restricta and M. globosa also decreased (Figures 1F and 1G).”

Accompanying this correction, we have also edited the legend to Figure 1 as follows (lines 205-208):

“The degree of skin colonization by overall Malassezia, M. restricta, and M. globosa decreased compared to baseline.”

Question 12

Figure 1

The letter "L" is written everywhere using bold - correction needed.

Response:

Thank you for pointing this out. The legend to Figure 2 may have appeared bold due to inconsistencies in font size (9 pt and 10 pt). Following your comment, we have adjusted the font size for consistency.

Question 13

Line 193, Figure 1

Should be “Candida albicans”

Response:

We have corrected “Candia albicans” to “Candida albicans” according to your indication (line 202).

Question 14

Lines 193-194, Figure 1

According to what is shown on Figure 1 "Relative abundance of Candia albicans" should be assigned as "d" and "Shannon diversity of the bacterial microbiome" should be assigned as "c". There is still a question to be clarified, namely whether it should be "Shannon diversity of the bacterial microbiome" or "Shannon diversity of the fungal microbiome"

Response:

We apologize for this error. The descriptions for Figure 1(c) and (d) were indeed correct, but figures were accidentally switched. This has been corrected in the revised manuscript. Moreover, the title of Figure 1 was about the fungal microbiome. This has also been corrected:

“Figure 1. Changes in the fungal microbiome after dupilumab administration.”

Question 15

Line 204

Why did for fungi Authors use the criterion "relative abundance greater than 2%" whereas for bacteria "relative abundance greater than 3%"? I think it should be explained in the text (maybe within Materials and methods)

Response:

Thank you for bringing this to our attention. This was an error we overlooked. In Figure 2, it was stated as, “…relative abundance of >3%..”, but we have now corrected this to, “…relative abundance of >2%...” in the revised manuscript. (Line 199,2360

Question 16

Line 206-208

It should be better in the substantive context if the Authors determined which specific species of the Staphylococcus genus were dominant and the decrease of which was observed after 12 weeks. There is a significant difference between the presence of S. aureus and S. epidermidis on the skin. The latter is more typical colonizer and component of the human skin microbiome. Do Authors have such data to refer to them more precisely here. This would be very valuable information.

Response:

Thank you for this valuable comment. We agree with your suggestion on this. Differentiating between S. aureus and coagulase-negative staphylococci (CNS) such as S. epidermidis would indeed provide useful information; but unfortunately, we were unable to identify all sequences assigned to the genus Staphylococcus to their respective species. Therefore, we have grouped them together as the genus Staphylococcus. Instead, the degree of skin colonization by S. aureus was quantified using qPCR.

Question 17

Lines 218-222

Authors wrote (quote) “The decrease in the relative abundance of the genus Staphylococcus following dupilumab administration….”

The interesting results obtained could have a better impact for the audience if in the "Introduction" or “Discussion” the Authors referred to the antibacterial activity of Dupilumab. Has anyone tested the antibacterial activity of this compound so far? Is there any information about it in the literature? Is there any information in the literature about the inhibitory activity of this compound (used solely or in combination) toward Staphylococcus aureus? In my opinion, this could be valuable information that would show in a better light the essence of the interesting results obtained by Authors. In my opinion, it would be good if the Authors referred in the "Introduction" or in the "Discussion" to the following articles (doi:10.1016/j.jid.2019.05.024; doi: 10.1111/j.1365-2133.2006.07410.x; doi: 10.1016/j.jaci.2018.08.022). All these papers deal with the effect of Dupilumab on Staphylococcus.

Response:

We thank the reviewer for this question and the suggestions. Our response to this question is similar to that for Question 9. We have not investigated the direct antibacterial effect of dupilumab on Staphylococcus, but because dupilumab is a recombinant human IgG4 monoclonal antibody against the α subunit of the human interleukin-4 receptor, it is reasonable to assume that it does not exhibit antibacterial activity.

The administration of dupilumab and topical steroids changes the pH of the skin lesion area to slightly acidic as symptoms improve. Staphylococcus aureus prefers a neutral pH environment for its growth, while coagulase-negative staphylococci (CNS) do not. In a weak acidic environment (pH 5.5), S. aureus cannot proliferate and CNS becomes dominant. In other words, dysbiosis occurs in the lesions of patients with atopic dermatitis. This explains why, after treatment, the colonization of S. aureus on the skin decreases and species diversity increases.

Following your suggestions in Question 9 and Question 17, we have added the following sentences in the “Introduction” section, along with your suggested references:

“This indicates that the change in skin environment—increase in moisture content and shift to a weak acidic pH—that accompanies the improvement in symptoms of AD results in the correction of skin microbiome dysbiosis. In addition to dupilumab, dysbiosis can also be treated with topical steroids [13].” (lines 57-61)

 “Furthermore, the severity of AD is correlated with the degree of skin colonization by Malassezia [18,19]; degree of Malassezia colonization on the skin of patients with moderate and severe cases is higher than in those with mild cases.” (lines 70-72)

The following reference has already been cited:

  1. Callewaert C, Nakatsuji T, Knight R, Kosciolek T, Vrbanac A, Kotol P, Ardeleanu M, Hultsch T, Guttman-Yassky E, Bissonnette R, Silverberg JI, Krueger J, Menter A, Graham NMH, Pirozzi G, Hamilton JD, Gallo RL.J IL-4Rα Blockade by Dupilumab Decreases Staphylococcus aureus Colonization and Increases Microbial Diversity in Atopic Dermatitis. J. Invest Dermatol. 2020, 140(1), 191-202.e7.

The following reference has been added in the revised manuscript:

  1. Gong, J.Q.; Lin, L.; Lin, T.; Hao, F.; Zeng, F.Q.; Bi, Z.G; Yi, D.; Zhao, B. Skin colonization by Staphylococcus aureus in patients with eczema and atopic dermatitis and relevant combined topical therapy: a double-blind multicentre randomized controlled trial. Br. J. Dermatol. 2006,155, 680-687. 

Question 18

Line 247

Instead of “Malassezia restricata” should be “Malassezia restricta”

Response:

We have corrected this typographical error according to your indication (line 255).

Question 19

Discussion

Lines 299-301

On the discussion pages, the Authors draw the following conclusion (I quote) “This indicates that improving the diversity of the skin bacterial microbiome and eliminating S. aureus could serve as microbiological markers indicating successful AD treatment”. Isn't this the Authors' conclusion too far-reaching and subjective? What are currently officially recognized criteria by dermatologists or organizations specializing in atopic dermatitis that could be recognized as microbiological markers indicating successful AD treatment. Do we currently have sufficient knowledge on this subject?

Response:

We appreciate the reviewer’s approach on raising this issue and understand that this statement might have been an over-reaching discussion. Therefore, to address the reviewer’s concern, we have deleted the following sentence from the revised manuscript:

“This indicates that improving the diversity of the skin bacterial microbiome and eliminating S. aureus could serve as microbiological markers indicating successful AD treatment.”

Question 20

Line 316

“...their primary source of carbon and nitrogen” sounds better and more completely

Response:

We have modified this sentence according to your suggestion (line 328):

“…their primary source of carbon and nitrogen [16].”

Question 21

Lines 317-319

I can agree that M. restricta and M. globosa are the most prevalent as the factors indirectly related with AD and seborrheic dermatitis, but I can't agree with the statement that these are the most prevalent etiological factors of pityriasis versicolor.

Response:

As indicated in the existing literature (references given below), Malassezia restricta and M. globosa are most prevalent in the lesion areas of patients with pityriasis versicolor. However, there are also reports stating that M. furfur is the most prevalent one. Therefore, to avoid confusion, we have removed the mention of pityriasis versicolor from our paper and revised the sentence as follows (line 331):

“….. but can cause seborrheic dermatitis, or folliculitis, and….”

  • Molecular epidemiology of Malasseziaglobosa and Malassezia restricta in Sudanese patients with pityriasis versicolor.

(Saad M, Sugita T, Saeed H, Ahmed A. Mycopathologia 2013, 175(1-2):69-74.) 

  • Molecular analysis of Malassezia microflora from patients withpityriasis versicolor.

(Morishita N, Sei Y, Sugita T. Mycopathologia 2006, 161(2):61-5.)

Question 22

Lines 319-320

On the discussion pages Authors wrote (I quote) “Following administration, the relative abundances of M. restricta and M. globosa decreased gradually”. How does this relate to the previous statement (I quote) and the results presented in Figure 1? “Furthermore, whereas the colonization levels of M. restricta increased by approximately 10% from 62.0 ± 11.0% at baseline to 72.0 ± 11.0%, the colonization levels of M. globosa decreased from 34.6 ± 8.4% to 24.6 ± 8.4% (Figures 1F and 1G)” – lines: 179-182. As I see there is discrepancy.

Response:

We apologize for this inconvenience. As you already pointed out in Question 11, the description in the “Results” section was incorrect. Therefore, we have made the following corrections and adjusted the figure legends accordingly. Please note that the description in the “Discussion” section is correct and will remain as is in the revised manuscript.

(line 186-193)

“The baseline degree of skin colonization by Malassezia was observed to decrease over time after administration of dupilumab, with reduction rates of 43.1 ± 30.7%, 49.2 ± 28.0%, 47.3 ± 31.7%, and 43.9 ± 29.2% at 2, 4, 8, and 12 weeks, respectively (Figure 1E). In addition, the degrees of skin colonization by M. restricta and M. globosa also decreased (Figures 1F and 1G).”

Question 23

Line 331

To the best of my knowledge Candida albicans can cause superficial infections of the skin, nails and mucous membranes not only in case of immunocompromised patients.

Response:

As you correctly noted, Candida albicans causes superficial infections. We have modified this sentence according to your suggestion (lines 353-354):

C. albicans can cause opportunistic infections in organs and mucous membranes of immunocompromised patients.”

Reviewer 3 Report

Comments and Suggestions for Authors

This study investigated changes in the skin microbiome after using dupilumab in AD patients, and it seems particularly meaningful that the fungal microbiome was studied well.

1. scale sample collection site

In the case of AD patients, was the sample taken from a lesion on the forehead? Are samples taken from the forehead of all patients even if there are no lesions? 

Also, were the healthy controls collected from the forehead area?

2. treatment methods

Is dupilumab only the treatment? Did authors perform other treatments in combination?

3. It would be good to explain in the discussion what mechanism authors think Malassezia decreased and diversity increased after using dupilumab.

Comments on the Quality of English Language

English language fine.

Author Response

Response to Reviewer’s comments:

Reviewer 3

This study investigated changes in the skin microbiome after using dupilumab in AD patients, and it seems particularly meaningful that the fungal microbiome was studied well.

Response:

We thank the reviewer for acknowledging our study and for the valuable feedback and suggestions. We have addressed each of your comments and made changes in the revised manuscript accordingly.

Question 1: scale sample collection site

In the case of AD patients, was the sample taken from a lesion on the forehead? Are samples taken from the forehead of all patients even if there are no lesions? 

Also, were the healthy controls collected from the forehead area?

Response:

We thank the reviewer for this question. In this study, scale samples were collected from the lesions on the forehead of all patients, as well as from the forehead area of healthy controls. To clarify this to the reader, we have added this information in the “Materials and Methods” section of the revised manuscript (lines 106-110):

“Skin scales were collected from the lesion site on foreheads using OPSITE FLEXIGRID Transparent Film Dressing (6 × 7 cm; Smith and Nephew Medical Ltd., Hull, UK), following the method described by Sugita et al. [27]. After collection, dressings were refrigerated to extract genomic microbial DNA. Skin scales were also collected from the forehead area of healthy controls.”

Question 2: treatment methods

Is dupilumab only the treatment? Did authors perform other treatments in combination?

Response:

In this study, we used not only dupilumab but also topical steroids and/or antihistamines. To clarify this, we have added the following text in the revised manuscript (line 101):

“Topical steroids and anti-histamine agents were consistently used during the study.”

Question 3:

It would be good to explain in the discussion what mechanism authors think Malassezia decreased and diversity increased after using dupilumab.

Response:

We thank the reviewer for this suggestion. Previous studies have shown that administration of dupilumab results in a decrease in skin colonization by S. aureus—an exacerbating factor for atopic dermatitis—and an increase in diversity of the bacterial microbiome. This is assumed to be due to the change in skin pH from neutral to weak acidic, which is accompanied by the improvement in skin symptoms of the patients (normal skin pH is weak acidic). In other words, S. aureus cannot survive in a weak acidic environment, but coagulase-negative streptococci (CNS) prefer this environment; therefore, in non-lesional sites of AD patients or the skin of healthy individuals, CNS becomes dominant. In the fungal microbiome, administration of dupilumab results in a decrease in skin colonization by Malassezia, and consequently, an increase in the diversity of the fungal microbiome was observed. This is a novel finding by our team. Unlike S. aureus, Malassezia can grow in both neutral and weak acidic environments, but inherently prefer a neutral environment. Therefore, it is believed that Malassezia is more prevalent in the lesion areas of AD patients at baseline than post-treatment with dupilumab.

We have added these discussions in the revised manuscript, according to your suggestion (lines 339-350:

“The mechanism for the decrease in skin colonization by Malassezia and increase in diversity of the fungal microbiome following administration of dupilumab is likely similar to that observed in the bacterial microbiome. As symptoms improve, the skin’s pH shifts from neutral to weak acidic. While S. aureus prefers a neutral environment, coagulase-negative Streptococci (CNS) have reduced growth in this environment, leading to dysbiosis [3]. On the contrary, S. aureus cannot grow in a weak acidic environment, but CNS prefers this environment, thereby resulting in an increase in diversity of the bacterial microbiome. Although Malassezia can grow in both neutral and weak acidic environments, it prefers a neutral environment. The fact that the fungal microbiome of patients with AD becomes closer to that of healthy individuals following the administration of dupilumab suggests that fungi other than Malassezia also prefer a weak acidic environment.”